# ATF4-Dependent *NRF2* Transcriptional Regulation Promotes Antioxidant Protection during Endoplasmic Reticulum Stress

**DOI:** 10.3390/cancers12030569

**Published:** 2020-03-01

**Authors:** Carmen Sarcinelli, Helena Dragic, Marie Piecyk, Virginie Barbet, Cédric Duret, Audrey Barthelaix, Carole Ferraro-Peyret, Joelle Fauvre, Toufic Renno, Cédric Chaveroux, Serge N. Manié

**Affiliations:** 1Centre de Recherche en Cancérologie de Lyon, INSERM U1052, CNRS 5286, Centre Léon Bérard, Univ Lyon, Université Claude Bernard Lyon 1, 69373 Lyon, France; carmen.sarcinelli@gmail.com (C.S.); Helena.DRAGIC@lyon.unicancer.fr (H.D.); Marie.PIECYK@lyon.unicancer.fr (M.P.); virg.faure@free.fr (V.B.); cedric.duret@lyon.unicancer.fr (C.D.); CAROLE.FERRARO-PEYRET@adm.univ-lyon1.fr (C.F.-P.); joelle.fauvre@inserm.fr (J.F.); toufic.renno@lyon.unicancer.fr (T.R.); 2Institute for Regenerative Medicine and Biotherapy, 34295 Montpellier, France; audrey.barthelaix@inserm.fr; 3Hospices Civils de Lyon, Centre de pathologie Est, 69500 Bron, France; 4Inserm U1242, Université de Rennes, Centre de Lutte Contre le Cancer Eugène Marquis, 35042 Rennes, France

**Keywords:** NRF2, ROS, ER stress, PERK, ATF4

## Abstract

Endoplasmic reticulum (ER) stress generates reactive oxygen species (ROS) that induce apoptosis if left unabated. To limit oxidative insults, the ER stress PKR-like endoplasmic reticulum Kinase (PERK) has been reported to phosphorylate and activate nuclear factor erythroid 2-related factor 2 (NRF2). Here, we uncover an alternative mechanism for PERK-mediated NRF2 regulation in human cells that does not require direct phosphorylation. We show that the activation of the PERK pathway rapidly stimulates the expression of NRF2 through activating transcription factor 4 (ATF4). In addition, NRF2 activation is late and largely driven by reactive oxygen species (ROS) generated during late protein synthesis recovery, contributing to protecting against cell death. Thus, PERK-mediated NRF2 activation encompasses a PERK-ATF4-dependent control of NRF2 expression that contributes to the NRF2 protective response engaged during ER stress-induced ROS production.

## 1. Introduction

The endoplasmic reticulum (ER) orchestrates protein folding and export. This function is sensitive to cellular alterations in energy levels, redox status, or calcium homeostasis [1], resulting in accumulation of misfolded proteins in the ER lumen, a condition known as ER stress [2]. The detection and resolution of ER stress relies on three major ER-spanning transmembrane proteins, PERK, inositol requiring enzyme 1 (IRE1) and activating transcription factor 6 (ATF6), which trigger the unfolded protein response (UPR). The UPR induces global translational and transcriptional changes to improve the ER protein-folding capacity. However, if ER stress cannot be resolved, the UPR shifts from pro-survival to pro-apoptotic signaling pathways [2]. Several lines of evidence suggest a pivotal role for the PERK branch in the cellular outcome of ER stress [3]. Activated PERK phosphorylates the alpha subunit of eukaryotic translation initiation factor 2 (eIF2α) that reduces protein synthesis but paradoxically increases the translation of mRNAs such as that of ATF4. ATF4 in turn initiates a transcriptional program including the up-regulation of the transcription factor C/EBP homologous protein (CHOP). ATF4 and CHOP cooperatively stimulate the expression of genes encoding functions in protein synthesis, and the reactivated translation rate generates significant amounts of reactive oxygen species (ROS) that induce apoptosis [4,5]. As part of the mechanisms engaged to limit oxidative insults, PERK has been shown to directly phosphorylate the master regulator of the cellular antioxidant response nuclear factor erythroid 2-related factor 2 (NRF2), leading to NRF2/Kelch-like ECH-associated protein 1 (KEAP1) complex dissociation and NRF2 stabilization and activation [6]. In addition, IRE1 can also activate the NRF2 and the subsequent oxidative stress response [7], indicating that the molecular events governing NRF2 activity upon ER stress are multifaceted. Apart from the classical activation of NRF2 relying on the induced-dissociation of the NRF2/KEAP1 complex, evidence suggests that NRF2 regulation also occurs at the transcriptional level and that increased transcription promotes resistance to ROS [8,9,10]. Notably, chromatin immunoprecipitation (ChIP) sequencing data from human mammary cells have exposed the direct binding of ATF4 to the promoter of the *NRF2* gene (*NFE2L2*) upon ER stress [11], suggesting that PERK-mediated regulation of NRF2 might be more complex than previously anticipated. 

Herein, we present an alternative mechanism to the direct phosphorylation by PERK that contributes to the protective NRF2-induced antioxidant response upon PERK activation. 

## 2. Results 

### 2.1. ER Stress-Independent PERK Activation Recapitulates ROS Production and NRF2 Activation

In order to investigate PERK-ATF4-dependent regulation of the NRF2 oxidative stress response, which can also be activated by IRE1 [7], we exclusively triggered the PERK pathway uncoupled from the other arms of the UPR to provide a simplified cellular context. To do so, the human cell line NCI-H358 was engineered to stably express the chimeric Fv2E-PERK protein in which the cytosolic PERK kinase domain is fused to a dimerization domain (Fv2E) with high affinity for the drug AP20187 (AP) [12]. This system allows the selective activation of the PERK signaling pathway uncoupled from the IRE1 and ATF6 arms of the UPR. Treatment of these modified NCI-H358 cells with AP resulted in an activation of the Fv2E-PERK construct, as revealed by a phosphorylation-induced shift in the migration of the protein, the phosphorylation of eIF2α and the accumulation of ATF4 (Figure 1a). In contrast, the two other UPR markers p58IPK and BiP reflecting the activation of the IRE1 and ATF6 branches of ER stress, respectively [13], were not affected. Next, we used the surface sensing of translation (SUnSET) approach [14], to monitor the rate of protein synthesis following Fv2E-PERK activation. We observed the partial inhibition of protein translation between 3 and 6 h of AP treatment, and its recovery at 24 h (Figure 1b). In line with published results [4], the recovery phase was concomitant with an increase in intracellular ROS (Figure 1c). We then assessed NRF2 protein amounts. NRF2 abundance, which is maintained low in unstressed conditions through Kelch-like ECH-associated protein 1 (KEAP1)-mediated proteasomal degradation, increased as early as 6 h after Fv2E-PERK stimulation and peaked at 24 h (Figure 1d). Comparable results were obtained in response to classical ER stress inducers, namely tunicamycin and thapsigargin (Figure 1e,f). To assess if this increase in NRF2 proteins contributed to buffer AP-induced ROS, we silenced NRF2 expression using siRNA. While NRF2 knockdown did not affect intracellular ROS levels under basal growth conditions, loss of NRF2 consistently increased amounts of ROS following Fv2E-PERK activation (Figure 1g). Finally, Fv2E-PERK activation reduced cell viability, which was further increased upon NRF2 silencing, and these effects relied on ROS production, as they are alleviated by treatment with the ROS scavenger N-acetylcysteine (NAC) (Figure 1h).

Therefore, the activation of the artificial eIF2α kinase recapitulates ROS production and NRF2 activation under conditions in which other ER stress-induced signaling pathways were not activated.

### 2.2. ROS Generated during Protein Synthesis Recovery Contribute to NRF2 Activation 

We next investigated the mechanism(s) of NRF2 activation. Although accumulation of ATF4 and NRF2 occurred concomitantly upon AP treatment, induction of their respective transcriptional program was delayed. Indeed, ATF4-regulated genes such as *TRB3, GADD34, CHOP* were induced from 6 h of AP treatment onwards (Figure 2a), whereas induction of NRF2 canonical target genes such as *NQO1, HO1* and *GCLC*, was observed only at 24 h (Figure 2b). In this latter case, the slight but significant augmentation of HO1 expression observed at 6 h, is likely due to the rapid onset of ATF4 activity since this gene is also positively-regulated by ATF4 [15]. These results indicate that even though the amounts of both ATF4 and NRF2 proteins increase 6 h following PERK activation, detectable NRF2 activity is delayed by 24 h compared to that of ATF4. 

To assess whether this increase in NRF2 is regulated by the canonical KEAP1 complex, we monitored NRF2/KEAP1 complex disruption using the Neh2-luc reporter. In this construct, the Neh2 domain of NRF2 responsible for its interaction with KEAP1, was fused to firefly luciferase and the protein stability of the reporter directly depends on its interaction with endogenous KEAP1 [16]. Figure 2c shows that the increase in Neh2-luc luminescence was not detectable at 6 h, but occurred after 24 h of AP stimulation. This result corroborates the measurement of NRF2 target gene expression presented above. Notably, PERK-mediated phosphorylation of NRF2 on threonine residues after 6 h of AP stimulation was not detected (Appendix A). Rather, NRF2 activation was correlated with protein synthesis recovery and ROS production. Indeed, the rise in Neh2-luciferase signal was alleviated by NAC treatment, indicating that oxidative stress is implicated in NRF2/KEAP1 dissociation (Figure 2c). Consistently, accumulation of activated NRF2 in the nucleus was mainly observed after 24 h of AP stimulation (Appendix A), and NAC reduced both NRF2 nuclear translocation (Figure 2d) and the induction of NRF2 canonical target genes (Figure 2e). In line with these results, leucine deprivation that triggers the eIF2α-ATF4 axis independently of PERK [17], also induced a concomitant ROS production and NRF2 nuclear translocation (Appendix A). Of note, protein kinase C-mediated phosphorylation of NRF2 Ser40 was decreased upon AP treatment (Figure 2d), suggesting that this mechanism for NRF2 activation is not engaged. To rule out the possibility that ROS-mediated NRF2 activation is specific to ER stress-independent PERK activation and/or to NCI-H358 cells, we used tunicamycin that activates PERK in a context of ER stress. Tunicamycin-induced NRF2 target genes were also strongly reduced by NAC in NCI-H358 cells (Figure 2f) or in HBEC-3KT cells (Appendix A). 

Collectively, these data demonstrate that following PERK activation, ROS generated during protein synthesis recovery contribute to activating NRF2. Therefore, NRF2 activation by the PERK pathway also relies on a complementary and/or alternative mechanism to its direct phosphorylation by PERK. 

### 2.3. The PERK Pathway Induces a Rapid ATF4-Dependent NRF2 mRNA Increase

Given that ATF4 can bind to the promoter of the *NRF2* gene (*NFE2L2*) upon drug-induced ER stress [11], we next assessed whether the PERK pathway could also regulate NRF2 at the transcriptional level. We initially confirmed this hypothesis by showing that (i) tunicamycin induced a rapid increase in *NRF2* mRNA, (ii) Fv2E-PERK activation and leucine deprivation, resulted in a similar induction (Figure 3a–c). These findings, in particular the latter also strongly suggested that the mechanism(s) controlling *NRF2* mRNA increase is independent of PERK-mediated phosphorylation of NRF2. 

We then tested whether ATF4 may control NRF2 mRNA up-regulation. ATF4 silencing strongly reduced NRF2 mRNA expression (Figure 3d, left panel) and protein amount (Figure 3f) as early as 6 h after Fv2E-PERK activation. Functionality of the ATF4 knockdown was confirmed by the reduction of the ATF4 target gene *TRB3* (Figure 3d, right panel) and of ATF4 protein surge (Figure 3f). The rapid ATF4-dependant NRF2 mRNA up-regulation was recapitulated upon tunicamycin treatment of H358, HBEC-3KT or A549 cells (Figure 3e and Appendix A). Consistently, ATF4 silencing decreased NRF2 protein abundance following global ER stress upon exposure to tunicamycin or leucine deprivation (Figure 3g,h). These results suggested that ATF4 may directly control *NRF2* mRNA. Indeed, we identified a putative C/EBP-ATF Response Element (CARE) in the human *NRF2* gene promoter through which ATF4 triggers transcription of a subset of target genes (Figure 3i). Chromatin immunoprecipitation (ChIP) assay following 6 h of AP treatment showed an enrichment in ATF4 binding to this CARE, comparable to that of the *Trib3* promoter region (Figure 3i). In addition, Figure 3j shows that the loss of ATF4 significantly decreased mRNAs level of NRF2 target genes upon AP stimulation, underlying the relevance of ATF4-dependent regulation of NRF2. Together, these data demonstrate that the PERK pathway engages a rapid ATF4-dependent up-regulation of *NRF2* mRNA expression. 

Finally, to investigate the biological relevance of the increase in *NRF2* mRNA and protein amounts, we overexpressed NRF2 in Fv2E-PERK cells (Figure 3k) and triggered the PERK pathway activation using AP. Cells overexpressing NRF2 displayed an increased viability compared to the control counterparts (Luc) following PERK pathway activation (Figure 3l), indicating that higher basal levels of NRF2 render cells prone to respond to oxidative stress following activation of the PERK branch of the ER stress.

Therefore, in addition to the direct phosphorylation of NRF2, the PERK pathway can also control NRF2 amounts through an ATF4-dependent mechanism, which reinforces the cytoprotective effects of NRF2 against ROS (Figure 4). 

## 3. Discussion

ER stress-associated PERK activation has previously been reported to induce rapid and direct phosphorylation of NRF2 [6]. Although this mechanism results in a ROS-independent activation of NRF2 during ER stress, we report that ROS generated during late stages of PERK-dependent protein synthesis recovery, also contribute to triggering the cytoprotective NRF2 antioxidant program. In addition, PERK activation triggers an ATF4-dependent control of *NRF2* mRNA abundance that reinforces the cytoprotective function of NRF2. These results show that the PERK pathway can engage different mechanisms to activate NRF2 during ER stress. 

Lu and colleagues notably demonstrated that PERK-dependent changes in gene expression are mainly reliant on transcription factors downstream of eIF2α phosphorylation [12], which has raised questions about the physiological importance of transcription activation by alternative PERK substrates such as NRF2 [2]. Our findings may reconcile these apparently discrepant results since the different mechanisms of NRF2 activation are theoretically not mutually exclusive but rather complementary or alternative. 

The PERK pathway induced an ATF4-dependent *NRF2* mRNA increase and the binding of ATF4 to the CARE of *Nfe2l2* promoter region supporting a direct regulation of *NRF2* gene expression. Interestingly, no conserved CARE was found in the orthologous region of the murine *Nfe2l2* promoter and no ATF4 binding to this promoter has been described in mouse embryonic fibroblasts [4], suggesting a species-specific *NRF2* regulation by ATF4. Fv2E-PERK-mediated ATF4-dependent increase in NRF2 protein amounts occurred relatively rapidly but did not readily translate into NRF2 activation. Considering that at steady-state, the abundance of NRF2 is maintained at a level significantly lower than that of KEAP1 [18], we hypothesize that a proportion of KEAP1 may act as a floodgate trapping the newly produced NRF2. Only when significant amounts of ROS are generated during protein synthesis recovery, is KEAP1 massively released from NRF2 leading to a strong NRF2 activation (Figure 4). 

In the context of tumorigenesis, there has been a renewed interest in NRF2 as a driver of cancer progression and resistance to therapy, over the last years. Many cancer cells hijack ER stress signaling to promote progression [2], thus our findings suggest that ATF4-dependent NRF2 activation could be aiding in this process, therefore supporting the interest in targeting NRF2 in cancer treatment.

In summary, we show that in addition to the reported rapid and direct phosphorylation of NRF2 by PERK, the PERK-eIF2α-ATF4 pathway controls NRF2 expression in human cells, which is at least in part activated by upcoming ER stress-associated oxidative stimuli. This work broadens our understanding of how cells cope with oxidative stress generated during ER stress. 

## 4. Materials and Methods

### 4.1. Cell Culture and Treatments

NCI-H358 cells were grown in RPMI 1640 supplemented with 10% fetal bovine serum (FBS) and 1% *v*/*v* penicillin/streptomycin solution. Puromycine (2 µg/mL) was used for the selection of Fv2E-PERK NCI-H358 cells, generated in-house. A549 cells were cultured in DMEM high glucose supplemented with 10% fetal bovine serum (FBS), glutamax and 1% *v*/*v* penicillin/streptomycin and HBEC-3KT were maintained in KSFM supplemented with bovine pituitary extract and recombinant human EGF. The cells were maintained at 37 °C in a 5% CO_2_ incubator. 

To induce ER stress, cells were treated with 0.5 μg/mL tunicamycin (Sigma-Aldrich, ref: T7765, St Louis, MO, USA) or 30 nM thapsigargin (Sigma-Aldrich, ref: T9033, St Louis, MO, USA) added to the culture medium. To induce oxidative stress or NRF2-mediated response, cells were treated with 100 μM H_2_O_2_ (Millipore, ref: 107209). To activate the Fv2E-PERK fusion protein cells were treated with 0.2 nM AP20187 (Clonetech, ref: 635060, St Louis, MO, USA). An equivalent volume of DMSO was used for control experiments. Leucine starvation experiments were performed using a DMEM/Nutrient Mixture: F-12 (DMEM/F12) medium devoid of leucine, glutamine, lysine and methionine (Sigma-Aldrich, ref: D9785, St Louis, MO, USA), in which glutamine, lysine, methionine and 10% dialyzed serum were subsequently added. The control medium for these experiments was supplemented with leucine.

### 4.2. siRNA and Plasmid Transfections

NCI-H358 cells were transfected with ON-TARGETplus SMART pool human NFE2L2 siRNA (L-003755-00-0010), ON-TARGETplus non-targeting siRNA (D-001810-03-50) (Dharmacon, GE Healthcare) or human ATF4 siRNA (sc-35112) (Santa Cruz Biotechnology, Dallas, TX, USA). All siRNA transfections were performed using 50 nM siRNA and HiPerFect reagent (Qiagen, Venlo, Netherland), according to the manufacturer’s traditional transfection protocol, for 72 h followed by treatments. Cells were seeded either onto 6-well plates (25 × 10^4^ cells per well), onto 24-well plates (5 × 10^4^ cells per well), or 96-well plates (2 × 10^4^ cells per well). 

For NRF2 overexpression experiments, cells seeded onto 6-well plates were transfected with a pcDNA3 vector encoding the Flag-NRF2 (addgene #36971) or luciferase by using the Attractene reagent (Qiagen), according to the manufacturer’s transfection protocol. 

### 4.3. Cell Extracts and Western Blot Analysis

Whole cell extracts were prepared from cultured cells lysed in the radioimmunoprecipitation assay (RIPA) protein buffer containing proteases and phosphatases inhibitors (Roche, Basel, Switzerland), and obtained by centrifugation at 16,000× *g* for 15 min at 4 °C. 

Subcellular fractionation was performed as follows: cells were detached by trypsin-EDTA and resuspended in 10 mM HEPES, pH 8.0, containing 10 mM KCl, 1.5 mM MgCl_2_, 0.5 mM DTT and protease and phosphatase inhibitors, incubated on ice for 10 min, to be then lysed by adding 0.1% NP-40. Cytoplasmic fraction was obtained by centrifugation at 1500× *g* for 10 min at 4 °C. The pellet, representing the nuclear fraction, was dissolved in the RIPA protein buffer containing protease and phosphatase inhibitors, and incubated on ice for 30 min. The nuclear fraction was obtained by centrifugation at 1500× *g* for 30 min at 4 °C. Tubulin and Fibrillarin were used as loading control for purity of cytoplasmic and nuclear fractions, respectively.

Protein concentrations of the cellular extracts were determined using the DC Protein Assay (Bio-Rad). Equal amounts of proteins (20 μg) were separated by SDS-PAGE and then transferred onto nitrocellulose membranes (Biorad, Hercules, CA, USA). Membranes were incubated in the blocking buffer, 5% milk or Bovine Serum Albumin (BSA) in Tris-Buffered Saline/Tween 20 (TBST), for 1 h at room temperature, then incubated overnight at 4°C with the appropriate primary antibodies, diluted in TBST containing 5% milk or BSA. Membranes were washed three times with TBST, incubated for 1 h at room temperature with the appropriate secondary antibodies, diluted in TBST containing 5% milk, and again washed three times with TBST. Detection by enhanced chemiluminescence was performed using the Clarity western ECL substrate (Bio-Rad, Hercules, CA, USA). 

For NRF2-flag immunoprecipitation, cells were transfected with the vector encoding the Flag-NRF2. 30 h later, cells were treated with PERK activators for 6 h. Then, cells were collected and lysed with the IP buffer (Tris-HCl 20mM pH 7.4, NaCl 300 mM, EDTA 1mM, NP40 0.5%, glycerol 10% complemented with proteases and phosphatases inhibitors). Supernatants were incubated with the Flag M2 beads (Sigma-Aldrich) overnight at 4 °C. Then beads were washed three times with IP buffer and samples elution was performed in Laemmli buffer at 95 °C for 5min. 

The primary antibodies used were purchased either from Santa Cruz Biotechnology: ATF4 (sc-200 and sc-390063) and α-Tubulin (sc-23950); from Cell Signaling Technology: PERK (3192), eIF2α (9722), P-eIF2α (3398), CHOP (2895), P-Threonine (9381) and p58IPK (2940); from Abcam: NRF2 (ab62352), P-NRF2 Ser40 (ab76026) and Fibrillarin (ab4566); from BD Biosciences: BiP (610978); from Enzo life sciences: FKBP12 (ALX-210-142); or from Millipore: Puromycin clone 12D10 (MABE343). The HRP-conjugated secondary antibodies were supplied by Jackson Laboratories (anti-rabbit and anti-mouse antibodies). Western blot images are representative of three independent experiments Proteins bands quantification was performed using the ImageJ software (NIH, Bethesda, MD, USA). 

### 4.4. SUnSET Assay 

Rates of protein synthesis in NCI-H358 cultured cells were evaluated using the surface sensing of translation (SUnSET) method as previously described [14]. NCI-H358 cells were seeded onto 6-well plates (2 × 10^5^ cells per well) and treated with 0.2 nM AP20187 for the indicated times. Cells were then incubated with 5 μg/mL puromycin (Sigma-Aldrich P9620, St Louis, MO, USA), added directly into the medium for 15 min at 37 °C, 5% CO_2_, prior to be harvested and processed to prepare whole cell extracts in the RIPA buffer. The amount of puromycin incorporated into nascent peptides was then evaluated by Western blot analysis by using anti-puromycin antibody. 

### 4.5. Luciferase Assay

Luciferase assays were performed in NCI-H358 cells seeded onto 24 well-plates (12 × 10^4^ cells per well), transiently co-transfected with Neh2-luc reporter plasmid [16] (0.4 μg for each sample) pRL-SV40 internal control vector (used at ratio of 10:1) and Attractene reagent (Qiagen), according to the manufacturer’s transfection protocol. Cells were collected in 100 µL of passive lysis buffer and analyzed for Firefly and Renilla luciferase activities with the Dual-Luciferase Reporter Assay System (Promega, ref: E1910), according to the manufacturer’s instructions. Luciferase activities were measured with a microplate reader (Tecan, Infinite M200 PRO, Life Science). The assays were repeated at least three times in triplicate.

### 4.6. Cell Viability Assay

Cell viability was determined using the CellTiter-Blue assay (Promega ref: G8081). NCI-H358 cells were plated onto 24-wells plate (5 × 10^4^ cells per well) prior to the NRF2 knock-down and the appropriate treatment (5mM NAC treatment prior to AP). CellTiter-Blue reagent was added to each well (100 µL per well), to the replaced medium (500 µL) and the plates were incubated at for 1 h 37 °C, 5% CO_2_. Fluorescence intensity (545 nm/600 nm excitation/emission) was measured using a microplate reader (Clariostar, BMG LABTECH). Cell viability is expressed as the mean percentage compared to the control. The assays were repeated at least three times in triplicate.

### 4.7. ROS Detection

Intracellular ROS measurement was performed using the fluorescent probe 5-(and-6)-chloromethyl-2′,7′-dichlorodihydrofluorescein diacetate acetyl ester (CM-H_2_DCFDA; C6827, Invitrogen Molecular Probes). Briefly, NCI-H358 cells were seeded onto 96-well plate (2 × 10^4^ cells per well). Cells were pre-incubated with 10 µM CM-H_2_DCFDA, diluted in PBS supplemented with 10% FBS, for 30 min at 37 °C, 5% CO_2_. Following the treatment, fluorescence intensity (490 nm/530 nm excitation/emission) was measured using a microplate reader (Clariostar, BMG LABTECH). A sister plate for each experiment was used to determine the cell number by CellTiter-Blue assay, and a calibration curve was obtained by plating an increasing the number of cells per well (2.5–40 × 10^3^). The assays were repeated at least three times in triplicate. The results were presented as the fold increase in ROS per cell. 

### 4.8. RNA Extraction and RT-qPCR 

Total cellular RNA was extracted using the NucleoSpin RNA Kit (Macherey-Nagel), according to the manufacturer’s protocol. For cDNA synthesis, 0.5 µg of RNA were reverse transcribed using Superscript II reverse transcriptase (Invitrogen, ref: 18064014, Carlsbad, CA, USA) with random primers (Invitrogen, ref: S0142, Carlsbad, CA, USA), according to the manufacturer’s instructions. cDNA was then amplified by qPCR using specific primers listed in Appendix A and the SYBR Green Master Mix (Biorad, Hercules, CA, USA). qPCR was performed using the CFX connect real-time PCR system (Biorad, Hercules, CA, USA). Expression of target genes was normalized against endogenous RPS11 mRNA levels, used as an internal control, and assessed using the comparative ΔΔCT method. qPCR experiments were repeated at least three times in triplicate. Primer list: *NQO1* Fw: GAAGAGCACTGATCGTACTGG, Rev: GGATACTGAAAGTTCGCAGGG, *HO1* Fw: TCTTCGCCCCTGTCTACTTC, Rev: CTTCACATAGCGCTGCATGG, *GCLC* Fw: CCTGTCTGGGGAGAAAGTTC, Rev: ACTCGGACATTGTTCCTCCG, *NRF2* Fw: ACATCGAGAGCCCAGTCTTC, Rev: AGCTCCTCCCAAACTTGCTC, *TRB3* Fw: TGGTACCCAGCTCCTCTACG, Rev: GACAAAGCGACACAGCTTGA, RPS11 Fw: AGCAGCCGACCATCTTTC, Rev: ATAGCCTCCTTGGGTGTCTTG, *GADD34* Fw: CTGTGATCGCTTCTGGCA Rev: GGAAGAAAGGGTGGGCATC and *CHOP* Fw: GGTATGAGGACCTGCAAGAGGT Rev: CTTGTGACCTCTGCTGGTTCTG.

### 4.9. Chromatin Immunoprecipitation (ChIP) Assay

Chromatin immunoprecipitation (ChIP) was performed using the SimpleChIP Plus Enzymatic Chromatin IP Kit (Magnetic Beads) (9005; Cell Signalling Technology, Danvers, MA, USA, Country). Briefly, NCI-H358 cells were seeded in 150 mm in diameter dishes (4 × 10^6^ cells per dish) and 3 dishes per treatment were used. Following the treatment with 0.2 nM AP20187 for 6 h, cells were washed with ice-cold PBS and cross-linked with 1% formaldehyde at room temperature for 10 min, to be then harvested in ice-cold PBS containing protease and phosphatase inhibitors. Chromatin was prepared and fragmented by partial digestion with Micrococcal Nuclease, followed by a mild sonication (Bioruptor Diagenode, Seraing, Belgium), according to the manufacturer’s protocol. Chromatin immunoprecipitations were performed using the ATF4 mouse monoclonal antibody (11815, Cell Signalling Technology), normal rabbit IgG antibody and ChIP-Grade Protein G Magnetic Beads both supplied by the kit. After reversal of protein-DNA cross-links, DNA was purified using spin columns. DNA enrichment was analyzed by qPCR using 1 µL of the template and the primers listed in Appendix A. The results were calculated as a signal relative to the total amount of input chromatin (ΔCT adjusted to % input), and fold-enrichment values are presented. Primer list: *Nfe2l2* Fw: GCTGAGCTTCCGAAAATCCC, Rev: GGGAGCTAACGGAGACCTC, *Trib3* Fw: GCGGATGCAGAGGAGAGA, Rev: CACTTCCGCTGCGAGTCT, Negative region Fw: CCCATGTCCCAGGAAGAAG, Rev: AGTCCTGGAAGGGGTAGTGG.

### 4.10. Statistical Analyses

Statistical analyses were performed using the GraphPad Prism 6 software (GraphPad Software, La Jolla, CA, USA) via one-way ANOVA (with Holm-Sidak’s post-test correction for multiple comparisons). All data are expressed as means ± SEM of at least three independent experiments, * *p* < 0.05, ** *p* < 0.01, *** *p* < 0.001.

## 5. Conclusions

How cells cope with harmful oxidative stress generated during ER stress is still not fully understood. It has been previously reported that PERK, a signaling molecule of the UPR, directly phosphorylates and activates the master regulator of the cellular antioxidant response NRF2 to limit oxidative insults. In this study, we show the existence of an alternative pathway in which PERK is involved in the regulation of NRF2. Upon ER stress, the activation of the PERK pathway promoted a rapid ATF4-dependent up-regulation of *NRF2* mRNA in human cells that contributed to the protection from ROS-associated cell death. Notably, the NRF2/KEAP1 protein complex dissociation, leading to NRF2 activation, was delayed and largely induced by ROS produced during protein synthesis recovery in late phases of the UPR. These findings broaden our understanding of how ROS-mediated cell death is counterbalanced during ER stress.

## Figures and Tables

**Figure 1 cancers-12-00569-f001:**
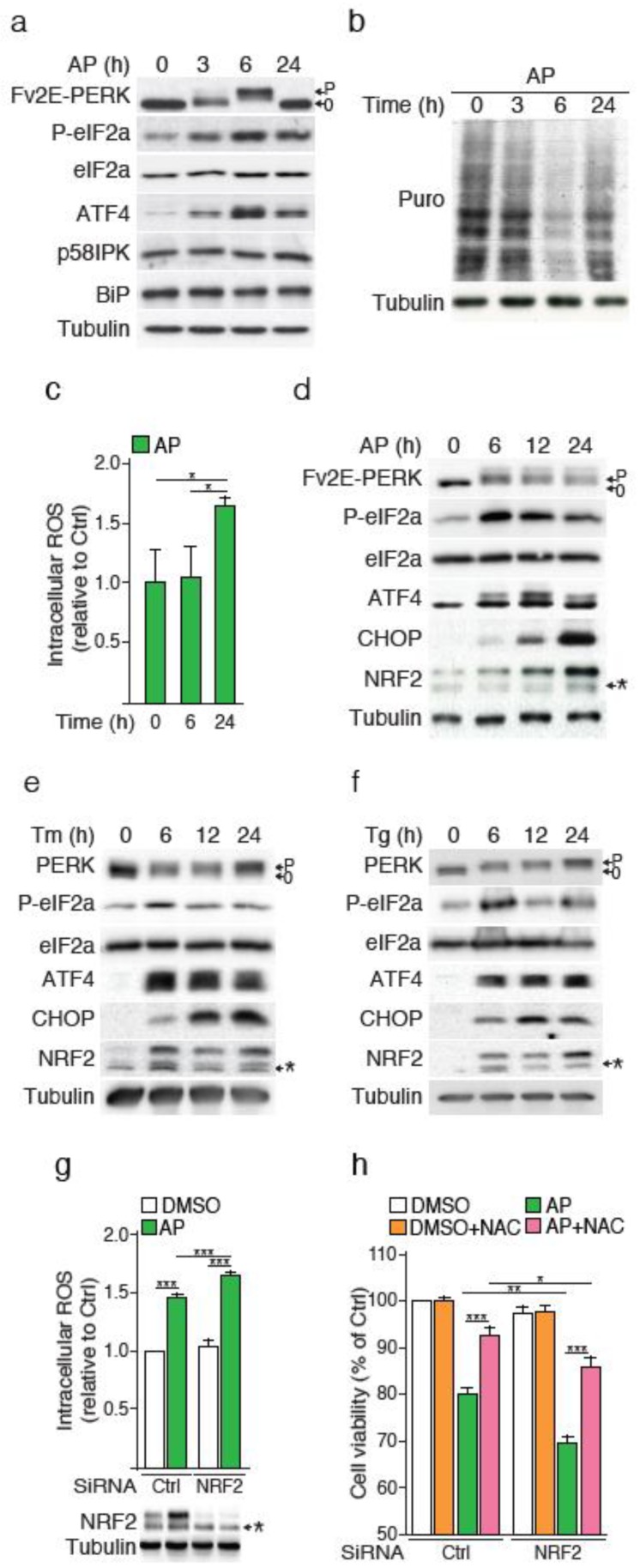
The sole activation of PERK recapitulates reactive oxygen species (ROS)-mediated cell death upon endoplasmic reticulum (ER) stress. (**a**) Time course analysis of Fv2E-PERK and unfolded protein response (UPR) markers following AP20187 (AP) treatment of NCI-H358 cells. Fv2E-PERK activation was assessed by monitoring changes in Fv2E-PERK band mobility upon AP treatment. P = phosphorylated and 0 = non-phosphorylated form of the protein. (**b**) Protein synthesis rate measurement by SUnSET assay. (**c**) Intracellular ROS measurement by fluorescence detection. Cells were treated with AP for 6 and 24 h. ROS fold increase per cell is reported. Time course analysis of Fv2E-PERK, P-eIF2a, ATF4, CHOP and NRF2following treatments with (**d**) AP, (**e**) tunicamycin or (**f**) thapsigargin. * = non-specific bands (**g**) Cell viability of NCI-H358 cells silenced for NRF2 upon PERK activation. NRF2 knockdown was confirmed by Western blot analysis. * = non-specific bands. (**h**) Cell viability of NCI-H358 cells silenced for NRF2 and treated with or without AP or N-Acetylcysteine (NAC). NAC was added to cell culture 1 h prior to adding AP for 24 h. Data are expressed as means ± SEM of at least three independent experiments, * *p* < 0.05, ** *p* < 0.01, *** *p* < 0.001.

**Figure 2 cancers-12-00569-f002:**
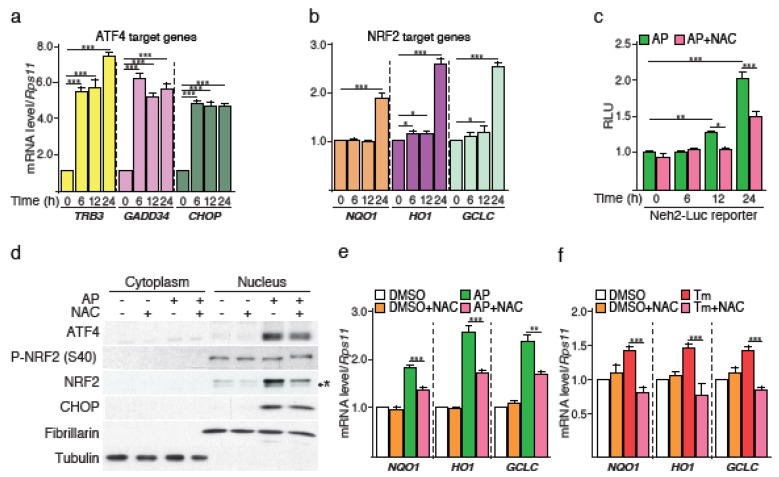
NRF2 activation by PERK partially relies on ROS generation during protein synthesis recovery. NCI-H358 cells were treated with AP over a time course of 24 h and RT-qPCR analysis were performed on canonical (**a**) ATF4-target genes (TRB3, GADD34, CHOP) or (**b**) NRF2-target genes (NQO1, HO1, GCLC). (**c**) Time course analysis of the KEAP1/NRF2 complex dissociation following AP treatment. CI-H358 cells were transiently co-transfected with the Neh2-Luc reporter and Renilla vectors and treated with NAC for 1 h prior to the addition of AP for the indicated periods of time. Data were normalized against the Renilla luciferase activity for each condition. Activity fold increase is reported. (**d**) Western blot analysis of ATF4, NRF2 and CHOP in cytoplasmic and nuclear fractions upon AP-mediated Fv2E-PERK activation with or without NAC. * = non-specific bands. (**e**) RT-qPCR analysis of NQO1, HO1 and GCLC expression levels. NCI-H358 cells were treated with or without NAC for 1 h prior to the addition of AP for 24 h. (**f**) RT-qPCR analysis of NQO1, HO1 and GCLC expression levels. NCI-H358 cells were treated with or without NAC for 1 h prior to the addition of tunicamycin for 24 h. Data are expressed as means ± SEM of at least three independent experiments, * *p* < 0.05, ** *p* < 0.01, *** *p* < 0.001.

**Figure 3 cancers-12-00569-f003:**
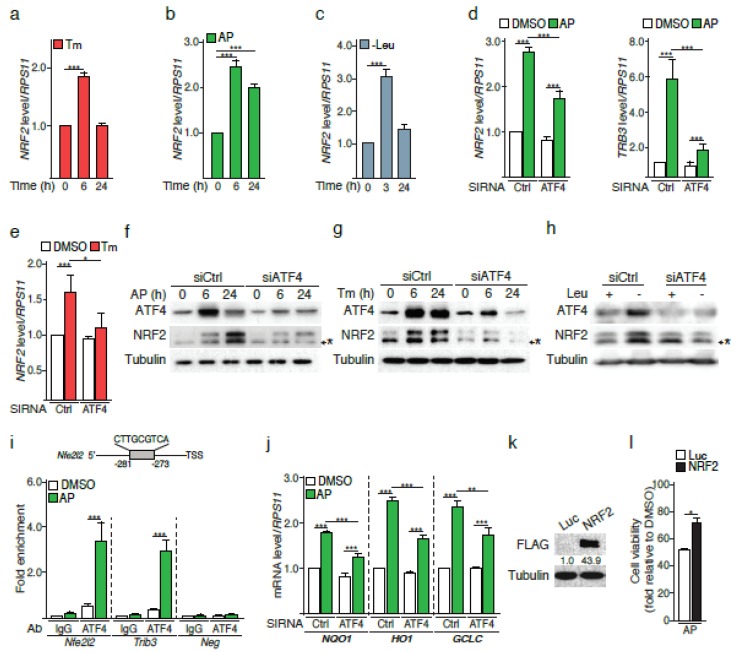
ATF4 directly controls NRF2 expression. Time course analysis of NRF2 mRNA levels in NCI-H358 cells upon (**a**) addition of tunicamycin (Tm), (**b**) AP treatment, or (**c**) leucine deprivation (-Leu). (**d**) *NRF2* and *TRB3* mRNA levels in NCI-H358 cells transfected with a siRNA Ctrl or directed against ATF4 and subjected to AP treatment for 6 h. (**e**) *NRF2* mRNA levels in NCI-H358 cells transfected with a siRNA Ctrl or directed against ATF4 and subjected to tunicamycin treatment for 6 h. Kinetics analysis of ATF4 and NRF2 amount from NCI-H358 cells transfected with a siRNA Ctrl or directed against ATF4 and treated (**f**) with AP or (**g**) tunicamycin. * = non-specific bands. (**h**) Immunoblot analysis of ATF4 and NRF2 amounts from NCI-H358 cells transfected with a siRNA Ctrl or directed against ATF4 and leucine-starved for 6 h. (**i**) Chromatin immunoprecipitation (ChIP) analysis of ATF4 binding to a putative CARE located within the human Nfe2l2 promoter (-281bp upstream the TSS, negative sense). NCI-H358 cells were treated with AP for 6 h. Trib3 and Neg primers were used as positive and negative controls, respectively. (**j**) RT-qPCR analysis of NQO1, HO1 and GCLC mRNA levels, following a 24-h activation of FV2E-PERK in ATF4 silenced NCI-H358 cells. (**k**) Immunoblot analysis of the Flag tag from NCI-H358 cells transfected with a pcDNA encoding luciferase (Luc) or Flag-NRF2. (**l**) Cell viability measurement of NCI-H358 cells overexpressing Luc or NRF2 and treated with AP for 24 h. Data are expressed as means ± SEM of at least three independent experiments, * *p* < 0.05, ** *p* < 0.01, *** *p* < 0.001.

**Figure 4 cancers-12-00569-f004:**
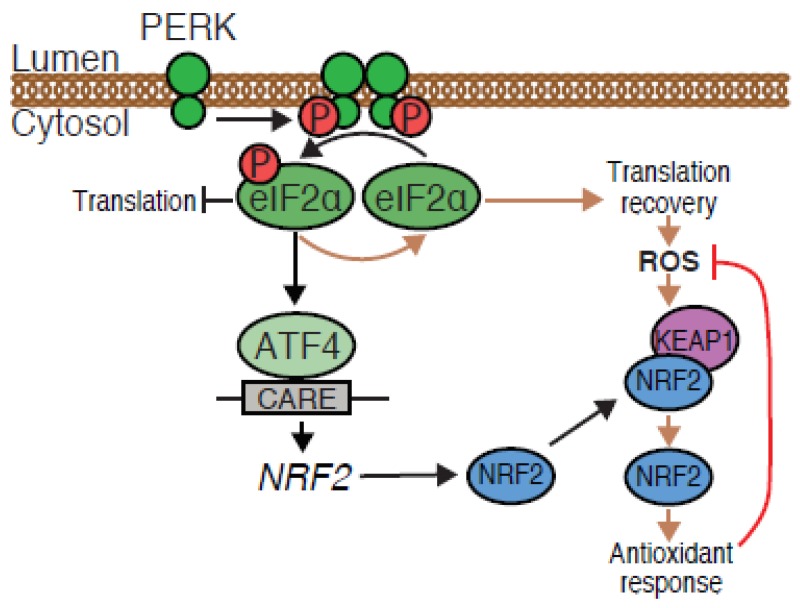
Schematic diagram of an alternative mechanism driving NRF2 activation upon PERK activation. In addition to the known NRF2 phosphorylation, activation of PERK triggers NRF2 transcription in an ATF4-dependent manner. During the protein synthesis recovery phase, subsequent generation of ROS induces the dissociation of the NRF2/KEAP1 complex, activation of the NRF2 antioxidant program and cell survival.

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
