# Peer review of "ATF4-Dependent NRF2 Transcriptional Regulation Promotes Antioxidant Protection during Endoplasmic Reticulum Stress"

_cancers, 2020, doi:10.3390/cancers12030569_

Round 1

Reviewer 1 Report

In this manuscript by  Sarcinelli et al,  the authors are reporting an alternative mechanism of PERK mediated NRF2 activation independent of the previously reported mechanism which is direct phosphorylation of NRF2 by PERK. Further they show that ATF4 activation by PERK is necessary to increase the mRNA levels of NRF2. Upon translational recovery, ROS production stabilizes NRF2 through dissociation from Keap1 to protect cells from ROS induced cell death. This independency of PERK phosphorylation is further demonstrated by the use of other stressors that activate eif2a-ATF4 axis independently of PERK which further strength their point.

Most of the data presented in this manuscript are of a good quality and the underlying logic is well communicated. These findings would be of great interest to the broad audience of cancers. Despite that several controls are missing and should be addressed to strengthen the overall conclusion of this manuscript: 

1) Earlier reports show PERK overexpression and dimerization cause NRF2 phosphorylation and nuclear translocation. I appreciate if the authors can show in their experimental conditions that even with NRF2 being phosphorylated the NRF2 is only translocated to the nucleus after translation recovery and ROS production.

2) In addition to the use of the chimeric Fv2E-PERK system which depends on PERK and its phosphorylation,  it will be more convincing to show ROS production and NRF2 nuclear translocation (rather than NRF2 protein levels) upon Leu deprivation, which in one hand induce ATF4 and also cause eIF2a phosphorylation independently of PERK phosphorylation.

3) It was shown previously that ROS stabilize NRF2 through oxidation of KEAP1-Cys-151 which in turn dissociate from Cul3 (an E3 ligase that is needed to targets NRF2 for degradation). This actually can be an alternative explanation why the authors still see the interaction between Keap1-NRF2 despite the stabilization of NRF2 at 6hrs in figure 2C which could be as a result of dissociation of Cul3-KEAP1.  In other words, can the author show the ROS effect on NRF2-Keap1-Cul3 complex in conditions that stabilize NRF2?

Minor points:

1) ATF4 appears sometimes as one band and in other gels as a double band in figure 1.

2) Figure 1A: eIF2a is phosphorylated at 3hrs but no phospho-PERK is observed at this time point. Can the authors show total eIF2a levels as well?

3) The kinetic of ATF4 induction seems to vary between experiments Fig1A it is highly induced at 6hours and then recovered at 24 hours while in Figure 1D those differences are vanished.  The same for phospho-PERK, in Fig1A it is fully phosphorylated at 6hours and then dephosphorylates at 24hrs a kinetics that is different from what is seen in figure 1D. Those are important controls given the claim that the NRF2 activation is depends on those factors. Phospho-eIF2a and total eIF2a protein levels are necessary in this case.

4) total and phospho eIF2a protein levels are needed in figure 1E and 1F as a control of translational inhibition, as mentioned earlier by the authors that IRE1 can cause activation of NRF2 independently of PERK.

5) Figure 1h lack the cell viability with DMSO only, how much cell killing the authors get with NRF2 silencing in these conditions?

6) Figure legends of Figure 1, FKBP12 was not introduced. Can the authors explain FKBP12.

7) Figure 2F is not called in the manuscript or mistakenly called as 2E.

Author Response

In this manuscript by Sarcinelli et al, the authors are reporting an alternative mechanism of PERK mediated NRF2 activation independent of the previously reported mechanism which is direct phosphorylation of NRF2 by PERK. Further they show that ATF4 activation by PERK is necessary to increase the mRNA levels of NRF2. Upon translational recovery, ROS production stabilizes NRF2 through dissociation from Keap1 to protect cells from ROS induced cell death. This independency of PERK phosphorylation is further demonstrated by the use of other stressors that activate eif2a-ATF4 axis independently of PERK which further strength their point.

Most of the data presented in this manuscript are of a good quality and the underlying logic is well communicated. These findings would be of great interest to the broad audience of cancers. Despite that several controls are missing and should be addressed to strengthen the overall conclusion of this manuscript:

1) Earlier reports show PERK overexpression and dimerization cause NRF2
phosphorylation and nuclear translocation. I appreciate if the authors can show in their experimental conditions that even with NRF2 being phosphorylated the NRF2 is only translocated to the nucleus after translation recovery and ROS production.

We concur with the reviewer and have added the analysis of NRF2 nuclear translocation at 6, 24 and 48h (Fig. S2) to show that it is mainly translocated at 24h when ROS production occurs. The text (line 140) has been modified to mentioned it.

 2) In addition to the use of the chimeric Fv2E-PERK system which depends on PERK and its phosphorylation, it will be more convincing to show ROS production and NRF2 nuclear translocation (rather than NRF2 protein levels) upon Leu deprivation, which in one hand induce ATF4 and also cause eIF2a phosphorylation independently of PERK phosphorylation.

We have performed experiments according to the reviewer’s suggestions. Fig.S3 shows that leucine induces ROS production together with a nuclear translocation of NRF2. These results have been included in the results section (line 142).

 3) It was shown previously that ROS stabilize NRF2 through oxidation of KEAP1-Cys-151 which in turn dissociate from Cul3 (an E3 ligase that is needed to targets NRF2 for degradation). This actually can be an alternative explanation why the authors still see the interaction between Keap1-NRF2 despite the stabilization of NRF2 at 6hrs in figure 2C which could be as a result of dissociation of Cul3-KEAP1.  In other words, can the author show the ROS effect on NRF2-Keap1-Cul3 complex in conditions that stabilize NRF2?

The reviewer raises an interesting possibility and we have attempted to monitor KEAP1 oxidation. Unfortunately, we could not clearly reveal KEAP1 by WB when following KEAP1 oxidation protocols. Obviously, we have been facing technical issues that could not be resolved within the timeframe allowed for the revision. Thus, whether KEAP1 oxidation can provide an explanation for NRF2 stabilization at 6h remains unclear. Although we will continue to investigate the possible mechanism underlying the early NRF2 protein stabilization, we believe that it is beyond the scope of this paper at this point in time.  

 Minor points:

1) ATF4 appears sometimes as one band and in other gels as a double band in figure 1.

During the course of the experiments, the company providing ATF4 antibody has changed its reagent. The novel antibody appears to have a different affinity for ATF4, which can be phosphorylated, and the antibody may recognize both forms of ATF4. Nevertheless, the ATF4 bands are specifics since their detection is abrogated upon siATF4.

 2) Figure 1A: eIF2a is phosphorylated at 3hrs but no phospho-PERK is observed at this time point. Can the authors show total eIF2a levels as well?

The total eIF2a level has been included in Fig. 1a.

 3) The kinetic of ATF4 induction seems to vary between experiments Fig1A it is highly induced at 6hours and then recovered at 24 hours while in Figure 1D those differences are vanished.  The same for phospho-PERK, in Fig1A it is fully phosphorylated at 6hours and then dephosphorylates at 24hrs a kinetics that is different from what is seen in figure 1D. Those are important controls given the claim that the NRF2 activation is depends on those factors. Phospho-eIF2a and total eIF2a protein levels are necessary in this case.

The kinetic of PERK de-phosphorylation upon AP treatment has proven to be variable. It was consistently activated already after 3h of AP treatment, but its de-activation/de-phosphorylation was less regular, being completely de-phosphorylated at 24h in some experiments while still phosphorylated in others. The reason for this is unclear. However, the kinetics of ATF4 induction were more consistent over the same experiments, reaching a maximum of induction between 6 and 12h, and decreasing after. ATF4 induction followed the level of the phosphorylation of eIF2a, which have been included in the Fig. 1d.

 4) total and phospho eIF2a protein levels are needed in figure 1E and 1F as a control of translational inhibition, as mentioned earlier by the authors that IRE1 can cause activation of NRF2 independently of PERK.

Total and phospho-eIF2a protein levels have been included in Fig. 1e and 1f.

 5) Figure 1h lack the cell viability with DMSO only, how much cell killing the authors get with NRF2 silencing in these conditions?

The silencing of NRF2 did not affect much the viability of H358 cells over the time course of the experiments. This is now showed in fig. 1h.

6) Figure legends of Figure 1, FKBP12 was not introduced. Can the authors explain FKBP12.

We thank the reviewer for noticing this typing error. FKBP12 should be read Fv2E-PERK instead. The dimerization domain Fv2E is derived form FKBP12 protein (Lu et al. ref 12) and the construct Fv2E-PERK is revealed using an anti-FKBP12 antibody described in the Mat. and Met. section. The mistake has been corrected in the legend of Figure 1.

 7) Figure 2F is not called in the manuscript or mistakenly called as 2E.

We thank the reviewer for noticing the mistake, indeed Fig. 2f is mistakenly called as 2e. This has been corrected.

Reviewer 2 Report

This is a comprehensive study to investigate direct transcriptional regulation of NRF2 expression by ATF4. The study was well carried out and the results were clearly presented. Although the results were strongly supported direct regulation of NRF2 expression by ATF4, there is still an issue about how NRF2 becomes transcriptional activation and upregulates its target gene expression in the absence of phosphorylated PERK (activated PERK), which is known to phosphorylate NRF2, thereby promoting dissociation from Keap1, and the phosphorylation of NRF2 is also required for its transcriptional activation. This concern was from the fig 1a showed no or low phosphorylation of PERK at 24 hr indicating that the PERK was inactive, while in fig 2b, the Nrf2 were actively up-regulated expression of its downstream target genes at 24 hr. Therefore, the results will be more complete if authors could include phospho-NRF2 results in figs 1a, e and f. This result will provide additional information about whether other kinases which could be activated by ROS, may involve in activation of NRF2 in the absence of PERK activity.    

Other comments

Phosphorylation of eIF2a only inhibits protein synthesis selectively, not global. Please change “phosphorylates the alpha subunit of eukaryotic translation initiation factor 38 2 (eIF2α) that reduces global protein synthesis” is line 38. In all figures’ legend, although it has been mentioned in M&M that all data were presented as mean±SEM, n=3. They still need to be written in the figure legend and also their corresponding p-value. Could authors explain why the activation of Fv2E-PERK is only transient in fig 1a? And why there was a reduction of PERK protein level upon phosphorylation, reduced over 80% at 6hr? Result of PERK in fig 1a looks quite different to fig 1d, although there was same experiment. The result was very clearly in fig 1a, but was not the case in fig 1d. In fig 3h, the label of + and – is the wrong way around?

Author Response

This is a comprehensive study to investigate direct transcriptional regulation of NRF2 expression by ATF4. The study was well carried out and the results were clearly presented. Although the results were strongly supported direct regulation of NRF2 expression by ATF4, there is still an issue about how NRF2 becomes transcriptional activation and upregulates its target gene expression in the absence of phosphorylated PERK (activated PERK), which is known to phosphorylate NRF2, thereby promoting dissociation from Keap1, and the phosphorylation of NRF2 is also required for its transcriptional activation. This concern was from the fig 1a showed no or low phosphorylation of PERK at 24 hr indicating that the PERK was inactive, while in fig 2b, the Nrf2 were actively up-regulated expression of its downstream target genes at 24 hr. Therefore, the results will be more complete if authors could include phospho-NRF2 results in figs 1a, e and f. This result will provide additional information about whether other kinases which could be activated by ROS, may involve in activation of NRF2 in the absence of PERK activity.    

To address the reviewer’s points, we have now investigated the phosphorylation of NRF2.

PERK phosphorylates NRF2 at threonine 80 (Bobrovnikova-Marjon et al. Oncogene. 2010 July 8; 29(27): 3881–3895). Because antibodies against phospho-Thr 80 of NRF2 are not available, we used a phospho-threonine reactive antibody, as described in the above publication. Figure S1 shows that we could not detect threonine phosphorylation of transfected and immunoprecipitated NRF2-flag construct upon AP or tunicamycin treatment. Thus, PERK-mediated phosphorylation of NRF2 may not be engaged in our cellular model. This is now mentioned in the text (line 136).

We also use an anti-phospho-Serine 40 that has been shown to be phosphorylated by the protein kinase C (Huang et al, J Biol Chem, 277 (2002), pp. 42769-42774), and which can be activated upon oxidative stress. Using the cellular lysates prepared for the results displayed in former Figure 2d, we now show that AP treatment decreases phosphorylation of this residue in Nrf2 (same apparent level of Ser40 phosphorylation, but the amount of NRF2 is increased in the nuclear fraction). This is now mentioned in the text (line 144).

Other comments

Phosphorylation of eIF2a only inhibits protein synthesis selectively, not global. Please change “phosphorylates the alpha subunit of eukaryotic translation initiation factor 38 2 (eIF2α) that reduces global protein synthesis” is line 38.

We agree with the reviewer comment, the word global has been removed from the text.

In all figures’ legend, although it has been mentioned in M&M that all data were presented as mean±SEM, n=3. They still need to be written in the figure legend and also their corresponding p-value.

This has been now included in the legends.

Could authors explain why the activation of Fv2E-PERK is only transient in fig 1a? And why there was a reduction of PERK protein level upon phosphorylation, reduced over 80% at 6hr? Result of PERK in fig 1a looks quite different to fig 1d, although there was same experiment. The result was very clearly in fig 1a, but was not the case in fig 1d.

As mentioned in response to reviewer #1, the kinetic of PERK de-phosphorylation upon AP treatment has proven to be variable. It was consistently activated already after 3h of AP treatment, but its de-activation/de-phosphorylation was less regular, being completely de-phosphorylated at 24h in some experiments while still phosphorylated in others. The reason for this is unclear at the moment. However, the kinetics of the phosphorylation of eIF2a and of ATF4 induction were more consistent over the same experiments, reaching a maximum of induction between 6 and 12h, and decreasing after. This is likely because the control of eIF2a de-phosphorylation is promoted by ATF4-regulated gene GADD34 which will oppose PERK-induced eIF2a phosphorylation.

The apparent reduction in PERK protein abundance mainly reflects a phosphorylation-induced shift toward higher molecular weights. As a consequence, the revealed bands are distributed over a larger gel area rather than being concentrated at the same MW, which contributes to the visual impression of a reduced abundance.

In fig 3h, the label of + and – is the wrong way around?

The eIF2a-ATF4 axis is activated upon the removal of leucine. Although the labelling may appear counterintuitive since the activated condition is usually labeled with a + mark, the actual labeling of figure 3h is correct.

Reviewer 3 Report

The authors report a novel mechanism for PERK-mediated NRF2 regulation via ATF4-dependent up-regulation of NRF2 mRNA thus leading to protection from ROS damage. This is an interesting article, well written, and with clear presentation of the research data; the experiments are well performed and substantiate the findings and interpretations.

A small suggestion: as the paper is submitted to "Cancers" it may be of interest to hypothesize in the Discussion how this new finding could potentially be exploited to further the field of basic or translational cancer research.

Line 206: replace "have" with "has"

Line 213: replace "is" with "are"

Please indicate what is the difference between the  various bands with differing Mr in NRF2 gels; a higher Mr intense band and two lower MR weaker bands. Does the asterisk in Figure 1e,f, 2d, 3f,g,h refer to one or both weaker bands?

A list of abbreviations is missing.

Author Response

The authors report a novel mechanism for PERK-mediated NRF2 regulation via ATF4-dependent up-regulation of NRF2 mRNA thus leading to protection from ROS damage. This is an interesting article, well written, and with clear presentation of the research data; the experiments are well performed and substantiate the findings and interpretations.

A small suggestion: as the paper is submitted to "Cancers" it may be of interest to hypothesize in the Discussion how this new finding could potentially be exploited to further the field of basic or translational cancer research.

A short paragraph has been added in the discussion, according to reviewer’s remark (line 238).

Line 206: replace "have" with "has". Line 213: replace "is" with "are"

We thank the reviewer for noticing the errors. They have been corrected.

Please indicate what is the difference between the various bands with differing Mr in NRF2 gels; a higher Mr intense band and two lower MR weaker bands. Does the asterisk in Figure 1e,f, 2d, 3f,g,h refer to one or both weaker bands?

The lower MW bands in NRF2 gels are non-specific bands recognized by the NRF2 antibody, because they are still detected upon siNRF2. The asterisk refers to both weaker bands and this is now indicated in all figures.

A list of abbreviations is missing.

A list of abbreviation is now provided at the end of the text.

Round 2

Reviewer 1 Report

No further comments for authors